

# Flow-Py v1.0: A customizable, open-source simulation tool to estimate runout and intensity of gravitational mass flows

Christopher J. L. D'Amboise[1,*], Michael Neuhauser[1,*], Michaela Teich[1], Andreas Huber[2],
Andreas Kofler[3], Frank Perzl[1], Reinhard Fromm[1], Karl Kleemayr[1,†], and Jan-Thomas Fischer[1,*]

[1]Austrian Research Centre for Forest (BFW), 6020 Innsbruck, Austria
[2]Unit of Hydraulic Engineering, Institute for Infrastructure Engineering, University of Innsbruck, 6020 Innsbruck, Austria
[3]Planungsgemeinschaft in.ge.na., 39100 Bozen, Italy
[*]These authors contributed equally to this work.
[†]deceased, February 2021

**Correspondence:** Christopher J. L. D'Amboise (Christopher.damboise@bfw.gv.at)

**Abstract.**

Models and simulation tools for gravitational mass flows (GMF) such as snow avalanches, rockfall, landslides and debris flows are important for research, education and practice. In addition to basic simulations and classic applications (e.g., hazard zone mapping), the importance and adaptability of GMF simulation tools for new and advanced applications (e.g., automatic

classification of terrain susceptible for GMF initiation or identification of forests with a protective function) are currently driving model developments. In principle, two types of modeling approaches exist: process-based physically motivated and data-based empirically motivated models. The choice for one or the other modeling approach depends on the addressed question, the availability of input data, the required accuracy of the simulation output, and the applied spatial scale. Here we present the computationally inexpensive open-source GMF simulation tool Flow-Py. Flow-Py's model equations are implemented via

the Python computer language and based on geometrical relations motivated by the classical data-based runout angle concepts and path routing in three-dimensional terrain. That is, Flow-Py employs a data-based modeling approach to identify process areas and corresponding intensities of GMFs by combining models for routing and stopping, which depend on local terrain and prior movement. The only required input data are a digital elevation model, the positions of starting zones and a minimum of four model parameters.

In addition to the major advantage that the open-source code is freely available for further model development, we illustrate and discuss Flow-Py's key advancements and simulation performance by means of three computational experiments:

1. Implementation and validation: We provide a well-organized and easily adaptable solver and present its application to GMFs on generic topograhies.

2. Performance: Flow-Py's performance and low computation time is demonstrated by applying the simulation tool to a

case study of snow avalanche modeling on a regional scale.





3. Modularity and expandability: The modular and adaptive Flow-Py development environment allows to access spatial information easily and consistently, which enables, e.g., back-tracking of GMF paths that interact with obstacles to their starting zones.

The aim of this contribution is to enable the reader to reproduce and understand the basic concepts of GMF modeling at the level of 1) derivation of model equations, and 2) their implementation in the Flow-Py code. Therefore, Flow-Py is an educational, innovative GMF simulation tool that can be applied for basic simulations but also for more sophisticated and custom applications such as identifying forests with a protective function or quantifying effects of forests on snow avalanches, rockfall, landslides and debris flows.

## 1 Introduction

The term gravitational mass flow (GMF) covers various natural hazard processes such as snow avalanches, rockfall, landslides or debris flows. GMFs are characterized by 1) the composition of their mass, and 2) the behavior of their motion (Köhler et al., 2018; Varnes, 1978; Okuda, 1991). However, certain commonalities are shared between most GMFs such as that their motion is driven by the force of gravity and that they are all processes acting on hill slopes (Varnes, 1978).

GMF simulation tools are crucial for developing natural hazard zoning maps and an integrated natural hazard risk management (Dorren et al., 2011; Guzzetti et al., 2002; Dorren, 2003; Barbolini et al., 2011; Sauermoser, 2006; Corominas et al., 2014; Fell et al., 2008; Fressard et al., 2014; Crozier and Glade, 2005; Guillard and Zezere, 2012; Van Westen et al., 2006). To optimize risk mitigation measures, e.g., by installing technical protection measures or planning and implementing nature-based solutions and avoidance strategies efficiently, GMF runout models can be used in economic studies (Fuchs et al., 2007; Moos et al., 2018; Teich and Bebi, 2009).

Many GMF specific models exist, which provide estimations of runout lengths for snow avalanches (Christen et al., 2010; Sampl and Granig, 2009; Christen et al., 2002; Lied and Bakkehøi, 1980; Bakkehøi et al., 1983; McClung and Lied, 1987), landslides (Brenning, 2005), or rockfall (Guzzetti et al., 2002; Dorren, 2012). More general GMF models can be applied to various GMFs and are either process-based physically (Sampl and Zwinger, 2004; Christen et al., 2010; Mergili et al., 2017; Wirbel et al., 2021) or data-based empirically motivated (Horton et al., 2013). The main differences between these two types are the larger number of input parameters and expensive computational resources required for process-based physically motivated GMF models (hereafter referred to as process-based models) in contrast to data-based empirically motivated models (hereafter referred to as data-based models) that usually involve less input parameters and are computationally inexpensive; however, process-based models provide more detailed information about a GMF process and its interactions with the terrain and obstacles in the flow path. The choice for one or the other modeling approach depends on the addressed question, the availability of input data, the required accuracy of the simulation output, and the applied spatial scale.

Depending on their application, one can choose between those two types of modeling approaches: process-based models are suitable for most applications provided that their input data requirements are met; however, to obtain detailed parameter sets over large areas is labor intensive and often not possible. Therefore, process-based models are best used on smaller (hill-





slope) scales and in data rich domains (Corominas et al., 2014; Van Westen et al., 2008), but methods to overcome the lack of
parameterizations have been developed tackling even back calculations, solving the inverse problem (Ancey et al., 2003; Eckert
et al., 2010; Fischer et al., 2015). In recent years, a number of data-based models, which require less input parameter have been
developed and applied to regional-scale case studies and for various GMFs. For example, random walk-based models have
already been applied to debris flows and other GMFs Gamma (1999); Mergili et al. (2015). Huggel et al. (2003) developed a
similar flow routing models and used it to assess GMFs related to glacier lake outbursts, but their model can also be applied
to other GMF types such as ice-rock avalanches (Huggel et al., 2007; Noetzli et al., 2006). Horton et al. (2013) published
the Flow-R simulation tool, which primarily aims at regionally assessing debris flow susceptibilities, but is also applicable to
other processes and variable friction relations. While data-based models mostly lack a physical interpretation of their results
they are computationally inexpensive and require less input data. In addition, data-based and process-based approaches can
be combined in one model (Scheidl and Rickenmann, 2011; Barbolini et al., 2011). Using a combination of observations, and
data-based and process-based models for hazard zone mapping has been proposed to overcome the lack of hard to measure
parameterizations for process-based models, especially for statistically sensitive variables (Barbolini et al., 2000).

We present the innovative and educational Flow-Py simulation tool, which employs a data-based motivated approach to
predict the magnitude, i.e., runout (spatial extent including starting, transit and runout zones) and intensity (effects of a GMF at
a specific location) of GMF processes. Flow-Py builds on the ideas and algorithms from existing data-based GMF models.The
Flow-Py algorithm is based on a flow path identification in three-dimensional terrain (routing) and concepts for runout and
intensity estimates along this path (stopping). To determine the GMF's runout and intensity we utilized well known runout
(travel) angle concepts (Heim, 1932), and derived corresponding geometrical quantities to motivate the Flow-Py model equa-
tions. These geometric relations serve further as reference to validate the Flow-Py implementation and results. In addition to
runout and intensity predictions, Flow-Py simulations results are also a measure of how exposed a location in the flow path is
regarding the number of starting zones and associated transit zones, which route flux through that location.

This contribution is structured as follows: In Sect. 2 we describe the motivation and implementation of our GMF model,
which is further explained in the code repository (Neuhauser et al., 2021). A validation experiment is presented in Sect. 3
which shows simulation results from three simple generic slopes. The performance of Flow-Py is tested via a regional scale
simulation of snow avalanches in Sect. 4. The customization of Flow-Py is described in Sect. 5 and shows how flexible the
simulation tool is and that it can be easily adapted with extensions to specific modeling questions.

With this contribution we enable the reader to reproduce and understand the basic GMF model concepts and their imple-
mentation in the Flow-Py code.

## 2 Model description

The main objectives of the Flow-Py simulation tool are to compute the spatial extent (hereafter referred to as runout) of GMFs,
which consists of starting, transit and runout zones, and the intensity of the GMF. Flow-Py is based on data-driven empirical
modeling ideas (Heim, 1932) with automated path identification (Holmgren, 1994; Horton et al., 2013; Huber et al., 2016;





Wichmann, 2017) to solve the routing and stopping of GMFs in three-dimensional terrain. Data-based models often require less input data, a less complex parameterization and solution (e.g. no time-dependent equations are usually solved) than process-based models. The Flow-Py simulation tool has been designed as a computationally inexpensive data-based model, which facilitates its application on regional scales, including a large number of GMF paths. Simulations of single starting cells take 1 to 10 seconds where process based, depth average simulations usually operate under the order of minutes. This can be attributed to the fact that no time depend equations, which process based models are built on, are solved in the underlying model equations of Flow-Py. The Flow-Py code is written in the Python computer language taking advantage of Pythons object-oriented class method. The well-structured model implementation allows users to address GMF specific modeling questions by keeping the parameterization flexible and enabling to include customized model extensions and add-ons. Flow-Py has already been applied to dry snow avalanches, rockfall and shallow-seated landslides by adapting the parameterization. Experience from similar studies also suggests that the model may also be suitable for other GMFs such as debris flows and wet snow avalanches Holmgren (1994); Gamma (1999).

The development philosophy to maximize the applicability of Flow-Py builds on:

1. flexible yet minimal input data requirements,

2. simple parameterizations which can describe a range of GMFs, and

3. a highly adaptable and customizable source code.

In the following sections the model motivation, implementation, input data and Flow-Py results, and underlying model equations are explained in detail.

## 2.1 Model motivation

The Flow-Py's routing and flow path identification in three-dimensional terrain was inspired by the gravitational process path model CPP (Wichmann, 2017), which introduced a weighting factor for the flow direction, and the programming architecture and persistence equations of Flow-R (Horton et al., 2013), combined with an adapted version of the flow direction algorithm (Holmgren, 1994) to appropriately model movement in flat and uphill terrain. The routing is based on local terrain and prior movement (flow direction and process intensity), which determines the flow path from starting to transit and runout zones and simultaneously describes the flow concentration, including lateral spreading. To estimate the process intensity along the identified path and the runout by introducing a stopping criterion we utilize the well-known runout angle ($\alpha$) concept (Heim, 1932; Lied and Bakkehøi, 1980; Bakkehøi et al., 1983; Körner, 1980) and derived corresponding geometrical quantities to motivate the Flow-Py model equations. Figure 1 depicts the runout angle along with the corresponding geometric relations in a two-dimensional representation along a GMF path, building the foundation for the underlying model equations.



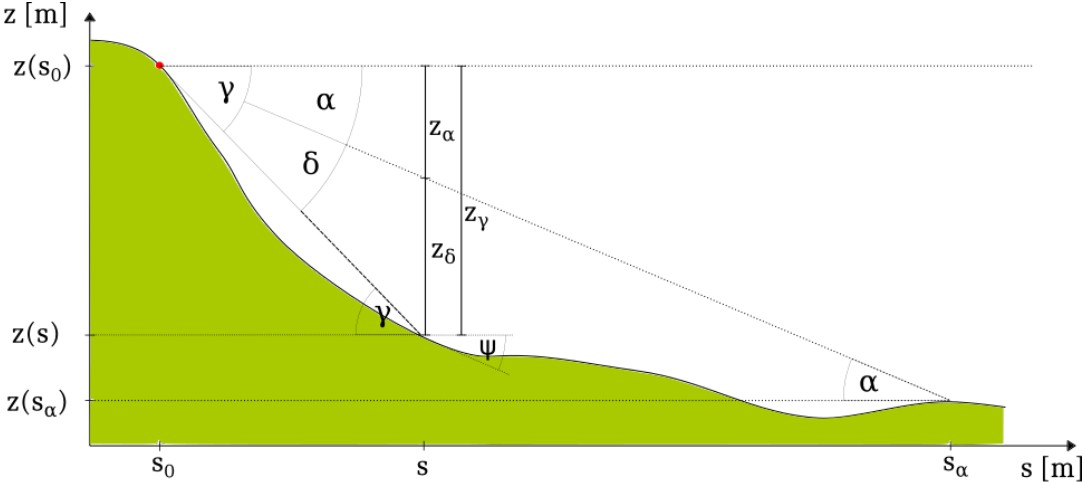

**Figure 1.** GMF path with altitude $z(s)$, projected travel distance $s$ and local slope angle $\psi$ with starting point $s_0, z(s_0)$ and runout point $s_\alpha, z(s_\alpha)$. The corresponding geometric quantities are directly related to the runout angle $\alpha$ concept and include the local travel angle $\gamma$, with corresponding total altitude change $z_\gamma$ and the process intensity measure $z_\delta$ with angle $\delta$.

The geometric relations are directly deduced from the runout angle $\alpha$ and allow to motivate the stopping and intensity estimates. Additionally, the geometric solution, represented by the $\alpha$-line from starting $(s_0, z(s_0))$ to the runout $(s_\alpha, z(s_\alpha))$ points, serves as reference for a model validation. Important quantities include the local travel angle $\gamma$:

$$\tan(\gamma) = \frac{z(s_0) - z(s)}{s - s_0}, \tag{1}$$

which, at the end of the GMF path, corresponds to the total travel angle (i.e., the so-called runout angle $\alpha$ (Heim, 1932)), that can be expressed as:

$$\tan(\alpha) = \frac{z(s_0) - z(s_\alpha)}{s_\alpha - s_0}. \tag{2}$$

The local travel angle height $z_\gamma$ corresponds geometrically to the total elevation drop $z_\gamma$ from the starting point $s_0$ to the currently projected runout length $s$ along the path:

$$z_\gamma = \tan(\gamma)(s - s_0)$$

$$= \frac{z(s_0) - z(s)}{s - s_0}(s - s_0) = z(s_0) - z(s). \tag{3}$$





The total elevation drop $z_\gamma$ splits into $z_\alpha$

$$z_\alpha = \tan(\alpha)(s - s_0)$$
$$= \frac{z(s_0) - z(s_\alpha)}{s_\alpha - s_0}(s - s_0), \tag{4}$$

which is associated with the dissipation kinetic energy height, and $z_\delta$

$$z_\delta = z_\gamma - z_\alpha$$
$$= z(s_0) - z(s) - \frac{z(s_0) - z(s_\alpha)}{s_\alpha - s_0}(s - s_0), \tag{5}$$

which is a measure of the process intensity, corresponding to the kinetic energy height (based on the principles of energy conservation, assuming a block movement with frictional dissipation associated to a Coulomb friction, Heim, 1932).

## 2.2 Implementation

The Flow-Py simulation tool is implemented based on object-orientated programming ideas, which allows for easy model customization (Neuhauser et al., 2021). Flow-Py is written in the freely available modern programming language Python3

(Van Rossum and Drake, 2009), which is widely used and supported by an active online community. The simulation tool is highly adaptable and different routing and stopping routines can be easily implemented, which enables the user to adjust the parameterization, also for multi model runs, and the equations that govern the movement of the mass down slope, and to implement Flow-Py in model chains. Flow-Py can be run either by command line allowing it to be called by external programs or in a BASH file, or with a simple GUI, which guides the user through choosing input files and the parameterization.


A GMF usually has one or more starting zones that span over a single or multiple starting cells. Flow-Py computes the so-called path, which we define as the spatial extent of the routing from each starting cell to the stopping cells. Each starting zone is associated with its own unique path; however, a certain location in the terrain can belong to many paths. Flow-Py identifies the path with spatial iterations on the cell level, starting with a single cell of a starting zone and then transferring the final

results of the cell and path levels to the output raster level (see Fig. 2). To route on the three-dimensional terrain operating on a quadrilateral grid, we implemented the geometric concepts that have been introduced in Sect. 2.1. That is, each path calculation starts with a starting cell, operating on the cell level, requiring the definition of parent, base, child and other neighbor cells (see Fig. 2). For the discretized model equations that operate on the cell level we use capital letters to distinguish the variables from the geometric motivation equations (see Sect. 2.1) with superscripts for the specification and subscripts for the cell indices.

The Python class object developed for Flow-Py is called Flow-Class, which can store values and functions. A Flow-Class is created for each cell that is part of one path when the neighbor cell is recognized as a child cell and is then added to the calculation queue. The Flow-Class saves information about a single cell, such as location, its parent cell(s), the output quantities, and other information needed for further calculations or computing the output raster. The cells in the calculation





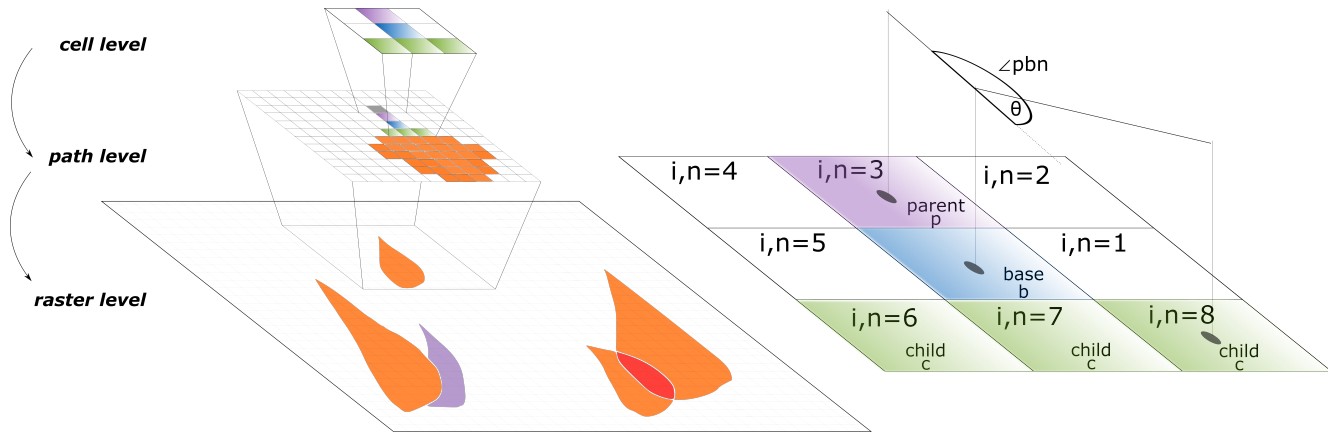

**Figure 2.** Left: The raster level summarizes the simulation results and output quantities for all GMF paths. The path level is the spatial level that contains the spatial extent of a path associated with one starting cell. Right: On the cell level, the iterative routing with parent $p$, base $b$, child $c$ and other neighbor cells $i, n$ is defined. The angle $\theta$ is the deviation between the projected incoming and outgoing directions, with the angle $\angle_{pbn}$ formed from the directions of parent $p$ to base $b$ and from base $b$ to neighbor $n$ cells.

queue will be the base cell (center cell) for subsequent calculations. Information on the iteration step is temporally stored to the

respective cell's Flow-Class. When the path calculations are finished, values from each cell's Flow-Class are updated to their respective location in the result array such that either a maximum value (e.g., $Z_{max}^{\delta}$, the maximum $Z^{\delta}$ for a cell over all path calculations) or a running sum (e.g., $Z_{sum}^{\delta}$, the sum of all $Z^{\delta}$ values for a cell over all path calculations) of all calculated paths that route flux through that cell are stored. The Flow-Class can be extended to store additional information that can be used to adjust stopping and routing calculations, e.g., the runout angle $\alpha$ is saved in the Flow-Class and could be adapted and scaled

with $Z_b^{\delta}$ to account for an energy dependent friction.

Using Python's object-oriented class method is a major advantage for advanced users since they can easily develop custom extensions or add-ons. We present an example for a back-tracking extension, which saves information of infrastructure located in GMF flow paths in the Flow-Class adapting the Flow-Py output in Sect. 5.

## 2.3 Input data

Flow-Py's core function loads and handles all input data, which are a digital elevation model (DEM) and a release raster in .asc or .tiff format. The release raster shows observed or potential GMF starting zones containing one or several starting cells. The release raster can be created by vector-to-raster conversion of polygon mappings by expert or by onset-susceptibility modelling. Flow-Py employs parallel processing for short model run times by splitting the release raster and DEM into tiles. Each tile is solved independently and sequentially in its own dedicated computer core and processing threads. Multi-processing is set as

the Flow-Py default, i.e. the number of free cores and the amount of RAM are first checked before splitting the starting cells





and spreading runout calculations among the free computer cores, making sure that the amount of RAM is not exceeded. The individual calculations are merged by updating the result arrays, which are transformed into output raster.

The release raster shows potential GMF starting zones containing one or several starting cells. The DEM and the release raster must be in the same extent and resolution with no resolution limit; however, 5 m and 10 m raster resolutions have been
tested. The major differences between different types of GMFs in regional runout modeling is the behavior of the movement and its runout, which can be summarized by the runout length and the convergence or divergence of the spreading movement. These behaviors are controlled by the parameterization of the stopping and routing routines in Flow-Py.

## 2.4    Model equations and path identification

The Flow-Py model equations are formulated with respect to an equidistant quadratic grid with the same resolution and extent
as the input raster. During each spatial iteration, calculations are made on a 3x3 cells subset of the raster, where the flux across the base cell (subscript $b$) is solved (see Fig. 2). The eight neighbor cells to the base cell (subscript $i$ and $n$) can be parent cells (subscript $p$) during an iteration step acting as flux source cells, or child cells (subscript $c$) acting as flux sink cells. The governing runout modeling question is broken down into two subquestions:

 1. Where does the GMF move to?

2. Where does the GMF stop?

These questions are addressed in two dedicated modeling routines called the routing routine and the stopping routine.

### 2.4.1    Routing

The routing routine considers a terrain contribution $T_i$ and a contribution accounting for prior motion called persistence $P_i$ (Horton et al., 2013); the flux is solved from parent cells through the base cell to child cells. Eq. (6) is the basis of the routing
algorithm and shows how the terrain contribution $T_i$ and the persistence contribution $P_i$ are combined to distribute the routing flux

$$R_i = \frac{T_i P_i}{\sum_{n=1}^{8} T_n P_n} R_b. \tag{6}$$

$R_i$ is the routing flux from the base cell to neighbor cell $i$ and $R_b$ is the total routing flux into the base cell (for starting cells $R_b = R_{start} = 1$). To conserve $R_i$, the amount of $R_b$ must be equal to $R_i$ unless a stopping criteria is met (see Sect. 2.4.4). To
conserve flux, $T_i$ and $P_i$ to cell $i$ are normalized across all neighboring cells $n$. The normalized direction is then scaled with $R_b$.

### 2.4.2    Terrain-based routing





The terrain-based routing accounts for the guiding effect of the slope on the movement. To distribute the flux we utilize the terrain routing function:

$$T_i = \frac{(\tan\phi_i)^{exp}}{\sum_{n=1}^{8}(\tan\phi_n)^{exp}} \; \forall \begin{cases} -90° < \phi_i < 90° \\ exp \in [1;+\infty] \end{cases} \tag{7}$$

where $T_i$ is the normalized terrain based routing from the base cell $i$ and $\phi_i = \frac{\psi_i + 90°}{2}$ is the distribution angle with the local slope angle $\psi_i$ from the center point of the base cell $b$ to the center points of neighbor cells $i$ where positive slopes indicate a downhill direction. The distribution function $\tan\psi_i$ is used as a weight to give preference for distributing flux to steeper slopes, where this distribution function allows for routing on flat and uphill terrain by returning values $< 0$ for $-90° < \psi_i < 90°$. The distribution function reaches a maximum at $\psi = 90°$, which is a vertical drop or free-fall, and a minimum at $\psi = -90°$ where $\tan\psi_i \approx 0$ occurring at a vertical rise or wall face.

To control the concentration of routing flux an approach based on the multiple flow direction algorithm for runoff has been employed (Holmgren, 1994). The exponent $exp$ together with the flux cutoff (see Sect. 2.4.6) controls the lateral spreading of the flow (Horton et al., 2013). When $exp$ increases, the terrain based routing flux is concentrated to the steepest decent. Together with the flux cutoff $> 0$, this results in the path's lateral spreading to be reduced. As $exp \to \infty$ the divergence results in a single flow direction (block movement) and as $exp \to 1$ wide spreading is encouraged (fluvial movement). However, other terrain-based routing approaches can be easily implemented in Flow-Py (see Horton et al. (2013) for summary).

### 2.4.3 Persistence-based routing

The persistence-based routing contribution aims to account for the influence or prior GMF movement on the subsequent routing. It must be noted that persistence is empirically derived and may be conceptually comparable to momentum; however, Flow-Py's underlying model equations do not account for mass (and hence momentum).

Equation (8) shows the persistence routing function $P_i$ for neighbor cell $i$:

$$P_i = \sum_{p=1}^{N_p} \sum_{n=1}^{8} Z_p^\delta D_n, \tag{8}$$

which is consists of two components, the direction $D_n$ Eq. (9) and the intensity $Z_p^\delta$, which has classically been called energy line height (Körner, 1980). Because a base cell can receive flux from many parent cells $p$ the persistence routing function is calculated over all neighbor cells $n$ considering the incoming flux from each parent cell.

The direction $D_n$ maintains the flow direction from a parent cell ($p$, flux source) to the base cell $b$. Weights are used to define the flow direction and are expressed as:

$$D_n = \max\{0, \cos(\theta)\}, \tag{9}$$





where $\theta = \angle_{pbn} - 180°$ is the resulting deviation angle between the projected incoming and outgoing direction, with $\angle_{pbn}$ as angle formed from the directions of parent $p$ to base $b$ and from base $b$ to neighbor $n$ cells, compare Fig. 2. Cells located opposite of a parent cell are assigned the full weight of one. Where cells 45∘ off of the direct flow direction get a weight of $\cos(45^°)$ or 0.707 similar to Horton et al. (2013). The reason that the persistence function passes flux through three cells and not only one is to compensate for the restriction that there are only eight directions to move on a raster grid. A weight of 0 is
given to all other cells including the parent cell.

The intensity $Z_p^\delta$ is stored in the Flow-Class of the parent cell $p$ from a previous iteration step, and the value of $Z^\delta$ is saved in the Flow-Class of each child cell. If one child cell has more than one parent cell, then $Z_{max,path}^\delta$ (maximum value of $Z^\delta$ for the many combinations of routes to a cell on a single path) is stored in its Flow-Class.

The intensity $Z_n^\delta$ at the neighbor cell $n$ is cell-wise calculated, i.e. cell to cell throughout the spatial iterations. The intensity
$Z_{bn}^\delta$ refers to the iterative part of $Z_n^\delta$ that is associated with the spatial step from the base cell $b$ to the neighbor cell $n$. Equation (10) shows the calculation of $Z_n^\delta$, where $Z_b^\delta$ is the intensity of the base cell $b$, which is stored in its Flow-Class. $Z_b^\delta$ was calculated on a previous spatial iteration when the current base cell was a child cell:

$$Z_n^\delta = Z_b^\delta + Z_{bn}^\delta, \tag{10}$$

where $Z_{bn}^\delta$ is calculated with respect to:

$$Z_{bn}^\alpha = S_{bn}\tan(\alpha), \tag{11}$$
$$Z_{bn}^\gamma = Z_b - Z_n, \tag{12}$$
$$Z_{bn}^\delta = Z_{bn}^\gamma - Z_{bn}^\alpha, \tag{13}$$

where the subscript $bn$ refers to base cell $b$ to neighbor cell $n$, with the distance $S_{bn}$ and the iterative energy quantities $Z_{bn}^\alpha, Z_{bn}^\gamma, Z_{bn}^\delta$ (see Fig. 1).
The total projected distance along the GMF path $S_n$ is expressed as

$$S_n = S_b + S_{bn}. \tag{14}$$

The parent cell further away from the stopping condition (larger $Z^\delta$) will have more influence on the routing flux. After all $n$ parent cells are calculated for each neighbor $i$ the persistence-based routing $P_i$ is combined with the terrain-based routing $T_i$, as seen in Eq. (7). When the parent cell has a large $Z_p^\delta$ the persistence-based routing $P_i$ will be the dominant term in Eq.
(7); however, if $Z_p^\delta$ is small, then the terrain based-routing $T_i$ will dominate the routing direction.

There are two limits that are imposed in the persistence routing routine: First, any cell that has previously been a base cell cannot be a child cell (a parent cell can not be a child cell). The disadvantage of this limit exerts on half-pipe shaped terrain in which the mass moves up a slope and back down on the same path but in the opposite direction. This limit is necessary to





keep small amounts of flux from routing back and forth in terrain shaped like a bowl. The major advantage of this limit is the
reduction of iteration steps by not calculating further flux for child cells resulting from flux oscillating in a bowl feature.

The second limit is imposed on the maximum value of $Z_i^\delta$, which is a limit of the process intensity ($Z_{lim}^\delta$), corresponding to a kinetic energy height or GMF velocity limit, respectively:

$$Z_n^\delta = \min(Z_n^\delta, Z_{lim}^\delta), \tag{15}$$

which is important for some GMF types, because it is analogous to introducing a turbulent friction coefficient in a process-
based model (Horton et al., 2013). In the examples used in Sect. 3, 4 and 5, no such limits are imposed.

### 2.4.4  Stopping

Two stopping criteria are employed: First, a runout angle criterion that limits how far the GMF runout goes. The second is a flux cutoff stopping routine, which, together with the divergence control ($exp$) in the routing routine, limits the lateral spreading of the path. The GMF will not propagate further, if either stopping criteria is met; however, the runout angle mainly determines
the total travel distance in the main flow direction, while the flux cutoff influences the lateral spreading.

### 2.4.5  Runout angle-induced stopping

The runout angle-induced stopping routine is based on the geometric quantities derived with the $\alpha$ angle concepts (c.f. Eq. (1 to 5); see Fig. 1). The local travel angle $\gamma_n$ is the inclination of the line formed from the top of the starting zone to the current neighbor cell $n$. The stopping condition is reached when $\gamma_n < \alpha$, i.e. when

$$Z_n^\delta < 0. \tag{16}$$

When the stopping condition is met, no child cells are assigned in the next iteration step.

### 2.4.6  Routing flux-induced stopping

The second stopping criterion is based on the assumption that a GMF must have a critical amount of routing flux $R_i$ to continue its propagation. If the GMF has an excessively divergent flow concentration that dilutes down and across a slope, then the flow
concentration (that can be associated to GMF mass) disappears at a critical amount of spreading, corresponding to the critical routing flux threshold $R_{stop}$.

The routing flux stopping criteria is met when

$$R_i < R_{stop}, \tag{17}$$





and the runout angle stopping condition is also met and neighbor cell $i$ is not a child cell. If $R_i \geq R_{stop}$ and the runout
angle stopping condition is not met, then neighbor cell $i$ is a potential child cell, and is added to the calculation queue and a
Flow-Class is accepted.

The routing flux-induced stopping mainly limits the width or spreading of the path. The magnitude of the routing flux of the
potential child cell $R_i$ relates to the percentage of initial routing flux from the start cell, where the starting flux $R_{start} = 1$. As
default $R_{stop} = 3 \cdot 10^{-4}$ has been adopted in Flow-Py and is shown in the examples in Sect. 3, 4 and 5.

**2.5  Flow-Py Outputs**

The outputs of Flow-Py are a set of raster in the same resolution and extent as the input DEM providing information about the
runout of the GMF and different measures of the intensity:

- $Z^\delta_{max}$ is the local maximum $Z^\delta$ for a cell over all path calculations. This is a geometric measure of highest intensity in
  terms of $Z^\delta$ for all starting cells which can be associated to maximum kinetic energy that is expected at each location
(raster cell).

- $R_{max}$ is the local maximum routing flux for a cell over all path calculations. This is a measure of intensity in terms of
  the maximum of flow concentration from a single start cell that is expected at each location (raster cell).

- $Z^\delta_{sum}$ is the sum of all $Z^\delta$ for a cell over all path calculations. This is a measure of intensity in terms of $Z^\delta$ combined
  with the number of starting cells that route flux through a location (raster cell).

- $CC$ is path cell counts, which is the number of paths that route flux through a location (raster cell). Together with $Z^\delta_{sum}$
  an average of $Z^\delta$ can be formed.

- $\gamma_{max}$ is the local maximum flow path travel angle for a cell over all path calculations. This is a measure of how exposed
  a location is with regards to how close the highest GMF intensity in terms of $Z^\delta$ is to the runout angle stopping criteria.

**3  Model testing and validation on generic slopes**

This first computational experiment demonstrates the Flow-Py routing and stopping algorithms for GMF modeling on simple
but increasingly complex generic topographies. We highlight how GMFs interact with different terrain features and show
the influence of different parameterizations on the flux; however, we do not perform a detailed parameter study, which is
beyond the scope of this contribution. First, we describe the scenarios (terrain and model parameterizations) and present the
simulations results. For each scenario, we altered the model parameterization or terrain complexity. Then the behavior of the
simulations and a comparison to the geometrically expected results, which allows for validation of the models implementation,
are summarized and discussed.

The generic topographies used for Flow-Py testing were generated using the generate topography functions provided within
AvaFrame (Wirbel et al., 2021). Terrain data was saved in ASCII raster format (.asc) with 10m resolution. The release raster





consisted of three 100 $m^2$-neighboring starting cells (starting zone = 300 $m^2$, 3 raster cells) located close to the top of the
generic terrain model at an elevation of 982 m. The three starting cells are centered on the y-plane.

### 3.1  Parabolic, open slope

The first example topography is built from a parabolic slope that connects with a flat (0° slope) plane. The extent of the terrain
model is 5000 m (x-axis) by 1500 m (y-axis). The transition from parabolic slope to flat plane takes place at 2250 m along the
x-axis. The total altitude difference of the terrain is 1000 m, with a maximum altitude located at x = 0. This parabolic slope
example is used as the base topography, to which more complex terrain features are added.

Figure 3 shows the parabolic slope and the results from two simulations, where the color scale is the $Z^\delta_{max}$ as an indication of
the intensity of the GMF. The parameterization used for these simulations are $\alpha = 25°$ , $R_{stop} = 3 \cdot 10^{-4}$ and $Z^\delta_{lim} = 8,849m$
(the height of Mount Everest, i.e. no effective limit is used)). The parameter that controls the concentration of flux ($exp$) is
varied between the two simulations to show results with low spreading (Fig. 3a, $exp = 100$) and high spreading (Fig. 3b,
$exp = 8$).

Comparing the top and bottom panels of Fig. 3, it can be seen that keeping the terrain, the runout angle and $R_{stop}$ the same
but reducing the $exp$ value, increases the spreading of the GMF, yet the runout length does not change. In the low spreading
example in Fig. 3a ($exp = 100$), the behavior of the downhill flow is restricted to a single flow direction in steeper terrain.
Once the slope flattens out the path diverges with very limited spreading. The small amount of spreading in flatter terrain can
be explained by the low $Z^\delta_{max}$, which results from the persistence-based routing being dominated by the terrain-based routing.
The front of the GMF runout is defined by the runout angle-induced stopping routine with $Z^\delta_{max} = 0$ (black). The sides of the
GMF process path are defined by the routing flux-induced stopping routine and because $Z^\delta_{max} > 0$ the runout angle-induced
stopping condition is not met.

### 3.2  Parabolic, channelized slope

This topography has the same extent, center line profile and configuration as the parabolic slope in Sect. 3.1; however, an
hour glass shaped channel is added, which begins wide and becomes narrow returning to a wide channel in the runout zone
again. The parameterization used for this scenario is $\alpha = 25°$, $exp = 8$, $R_{stop} = 3 \cdot 10^{-4}$ and $Z^\delta_{lim} = 8,849m$, such that one
can compare it to the simulation results shown in Fig. 3b.

This example highlights the routing flux-induced stopping and the terrain-based routing (Fig. 4). The GMF travels down
the channel and does not spread like in the previous example (Fig. 3). That is, the routing algorithms acts on the channelized
terrain and concentrates the flux in the center of the channel. The GMF does not spread outside of the channel because the flux
that is routed up the channels walls does not exceed the flux cutoff $R_{stop}$, hence the routing flux-induced stopping criteria is
met.





**Figure 3.** GMF runout modeled with Flow-Py on a simple parabolic slope connected to a flat plane with the runout angle $\alpha = 25°$. The divergence control is exemplified with a low spreading (a, $exp = 100$) and a high spreading (b, $exp = 8$) simulation. Both examples use a flux cutoff of $R_{stop} = 3 \cdot 10^{-4}$ and $Z^{\delta}_{lim} = 8,849m$ (the height of Mount Everest, i.e. no effective limit is used). Cooler colors indicate areas where the process has a relatively low intensity with regards to $Z^{\delta}_{max}$ and warmer colors show areas where the process has a relatively high intensity with regards to $Z^{\delta}_{max}$, which is associated to maximum kinetic energy..



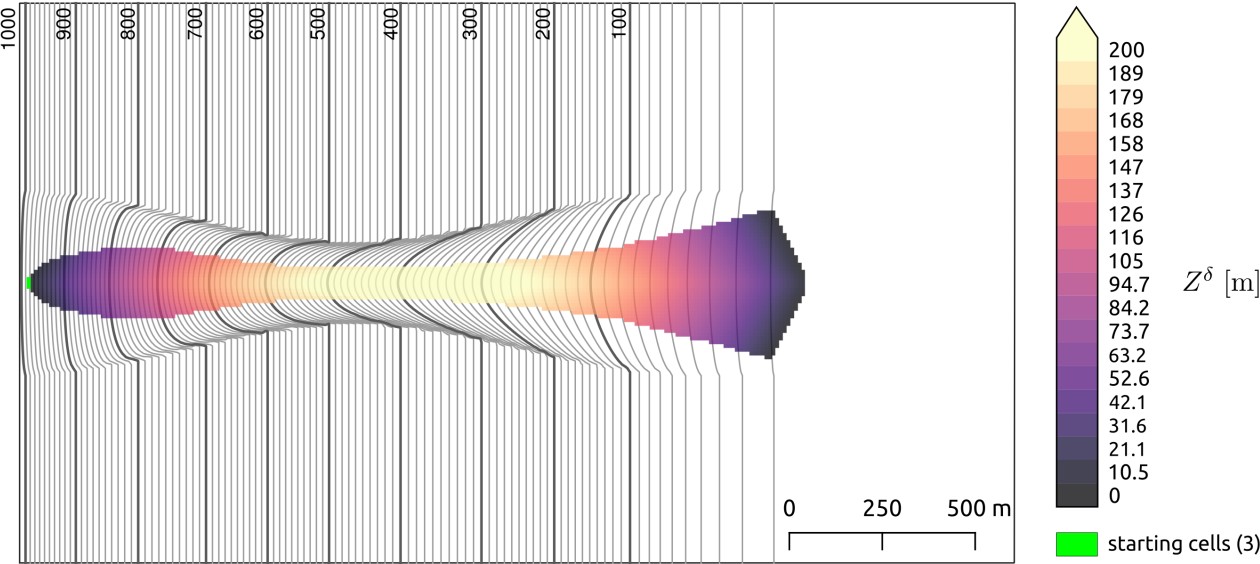

**Figure 4.** GMF runout modeled with Flow-Py on a parabolic slope with a channel with $\alpha = 25°$, $exp = 8$, $R_{stop} = 3 \cdot 10^{-4}$ and $Z^{\delta}_{lim} = 8,849m$. The colors show the value of $Z^{\delta}_{max}$, which is associated to maximum kinetic energy. The topography is a simple parabolic slope connected to a flat plane.

### 3.3 Parabolic, channelized slope with superimposed dam

The topography used in this scenario is the same as in the last Sect. (3.2) including a superimposed obstacle that crosses the terrain such that the GMF must travel uphill to overcome it. We refer to this obstacle as a dam as it could resemble a dam built in the GMF path. This example highlights how the Flow-Py simulation responds to flat or uphill terrain, which is where persistence-based routing will dominate over the terrain-based routing. The parameterization used is $\alpha = 25°$, $exp = 8$, $R_{stop} = 3 \cdot 10^{-4}$ and $Z^{\delta}_{lim} = 8,849m$, so that the result can be directly compared with the spreading example shown in Fig.

3b and the channelized example (Fig. 4). The dam has a shape of a Gaussian function with a width of 75 m and a height of 75 m which is added on top of the topography of the parabolic slope with a channel. The center of the dam (maximum height) is located at 1350 m (Fig. 5).

The GMF traveled just as far as in the previous examples, but its spreading increased once it encounters the dam since uphill terrain is more divergent (Fig. 5). The GMF has a lower $Z^{\delta}_{max}$ or energy when reaching the top of the dam; however, after the

dam the intensity is the same as in previous examples resulting in the same runout length with a slightly different lateral shape.

### 3.4 Discussion on model testing and validation

The Flow-Py simulation tool is based on a simple model that allows for regional application and was not specifically designed to model a singular GMF. However, simulations on generic topographies and of single paths provide a visual description of

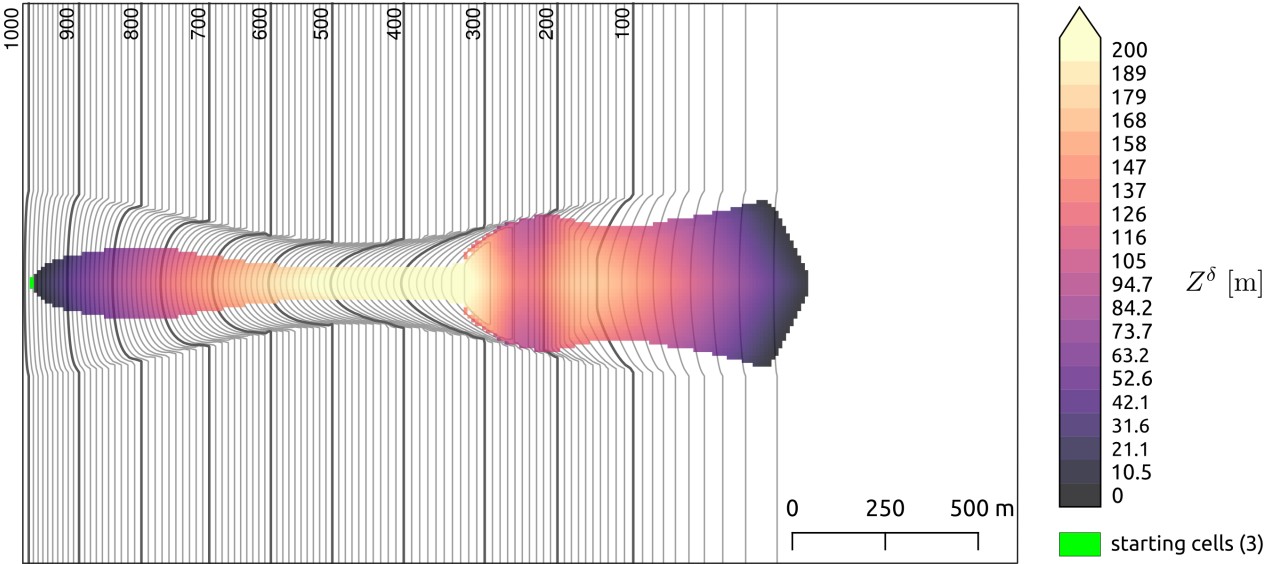

**Figure 5.** GMF runout modeled with Flow-Py on a parabolic slope with a channel and a dam that crosses the terrain at 1350 m with $\alpha = 25°$, $exp = 8$, $R_{stop} = 3 \cdot 10^{-4}$ and $Z_{lim}^{\delta} = 8,849m$. The colors show the value of $Z_{max}^{\delta}$ which is associated to maximum kinetic energy. The topography is a simple parabolic slope connected to a flat plane.

how the implemented routing and stopping routines react to different terrain features and parameterizations. The parameters

$R_{stop}$ and $exp$ are primarily responsible for limiting the spreading of the path, where $\alpha$ and $Z_{lim}^{\delta}$ are primarily responsible for limiting the runout distance. $R_{stop}$ and $exp$ are dependent on the resolution of the DEM, where $\alpha$ and $Z_{lim}^{\delta}$ are not.

Figure 6 shows $Z_{max}^{\delta}$ values for the center line of all scenarios presented in this Sect., which allows to quantify and validate the model implementation. All simulations yield the same values for $Z + Z_{max}^{\delta}$ (where Z is the terrain height) along the center line, although topographies and associated three-dimensional runout extents differ significantly. This is particularly interesting

for the third scenario (Fig. 5) where not only $Z + Z_{max}^{\delta}$ values are matched but the routing and propagation of the GMF continued beyond the obstacle, where it usually would prohibit any propagation, e.g., with an often employed steepest descent routing approach.

In addition, the model motivation allows to predict the geometrically and theoretically expected solution in terms of runout and $z_{\delta}$. By comparing the geometrically correct solution $z_{\delta}$ with the simulation results of $Z_{max}^{\delta}$ for each scenario we obtain a

match with root mean squared error of $4 \cdot 10^{-5}$ for each simulation result compared with the geometric solution (see Fig. 6). This in turn validates the discretized model equations and their correct implementation. That is, the cell by cell approach to the routing results in expected behavior with all the stopping points matching the geometric solution even on flat and uphill terrain with very high accuracy. Furthermore, $Z_{max}^{\delta}$ values solved on the 10 m grid for each scenario fit the continuous geometric solution. This validation however is only relevant for the intensity $Z_{max}^{\delta}$ and runout length along the center line. It was not

the aim to fully validate the implementation of the spreading algorithm; however, the scenarios show satisfying results where





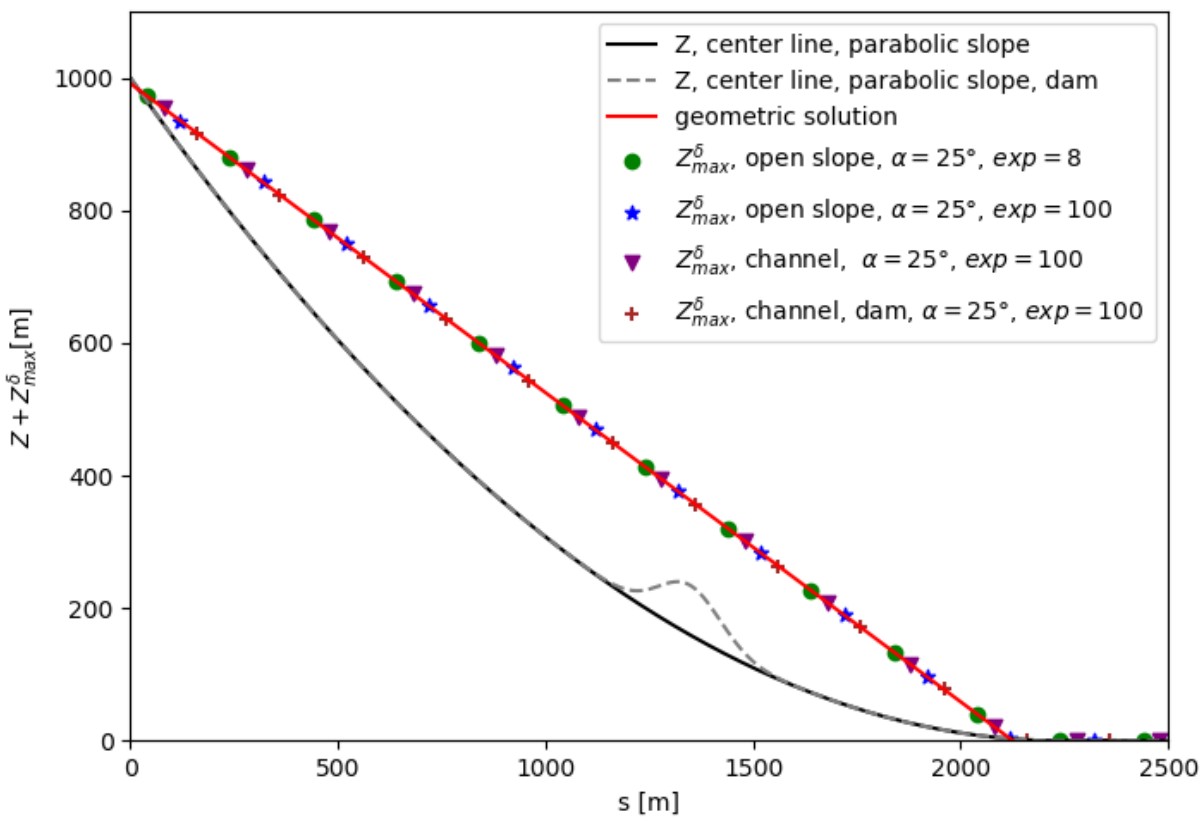

**Figure 6.** Geometric solution for the stopping criteria (red line) represented by the $\alpha$-line from starting to the runout points with $Z_{max}^{\delta}$ values for the center lines of scenarios presented in Sect. 3.1 and 3.2 (black line), and 3.3 (black dashed line).

single flow and divergent flow behavior, i.e. ranges from block to fluvial GMF behavior, can be reproduced by changing the Flow-Py parameterization (Fig. 3).

## 4   Performance testing on a regional scale

This section is dedicated to highlight the performance of the Flow-Py simulation tool in real terrain and on a regional scale by
applying it to the snow avalanche GMF.





### 4.1 Study area description and experimental setup

The study area is located in the mountains surrounding the Austrian villages Vals and Gries am Brenner in Tyrol close to the Italian border (Plörer and Stöhr). The area of the study area is 104.5 $km^2$. The input DEM is freely available from Land Tirol (data.tirol.gv.at) issued under a Creative Commons Attribution 4.0 International (CC BY 4.0) license.

The computation time is dependent on the number of starting cells and the extent of the paths (how divergent/concentrated). We developed an overly simple starting zone model to test the performance of Flow-Py on a regional scale. There are many models for identifying potential avalanche starting zones that use a range of slope inclinations such as 28 ° to 60 ° Veitinger et al. (2016); Maggioni and Gruber (2003); Pistocchi and Notarnicola (2013). More information such as terrain curvature, forest cover and average maximum snow depth are used to further restrict the number and size of potential starting zones. The

starting zone model employed is based solely on the slope inclinations derived from the 10 m DEM with the goal to provide a sufficient number of starting cells with potentially long runout lengths for performance testing. To achieve this we used two criteria for identifying starting cells: first, starting cells must be located above 1800 m, second, the starting cell must have a slope inclination between $31°$ to $34°$. The range of slope inclinations used is much smaller than used in more sophisticated models, this method was used to reduce the number of starting cells with out introducing more information such as forest area,

or average snow depth, but rather relying solely on the 10 m DEM.

The parameterization for this simulation was $\alpha = 25°$ and $exp = 8$ and $R_{stop} = 3 \cdot 10^{-4}$ and $Z_{lim}^\delta = 8,849m$, which have successfully been used to model large to very large avalanches (D'Amboise et al., 2021). For snow avalanches an $exp$ of 8 on a 10 m resolution DEM has produced good results in the past studies (Huber et al., 2016).

### 4.2 Results

The study area contains 1045311 raster cells 104531100 $m^2$) and starting cells comprise 5.4% of the total study area 56969 raster cells or 5696900 $m^2$ ), which can be seen in Fig. 7. The simulation took 3 h and 45 min with multi-processing on 16 cores.

Flow-Py identified 642630 cells or 61.5 % of the total study area as part of the avalanche starting, transit and runout zones (see Fig. 8 ). Many of the these cells belong to multiple paths and are, therefore, base cells for many calculations which is

reflected in the CC (cell counts) output raster. The CC output is not shown, however all the example input data and simulation results can be found in D'Amboise et al. (2021).

### 4.3 Discussion

The GMF path (extent of the avalanche starting, transit and runout zones) is determined by the length of runout and the amount of spreading. Because of the over simple starting zone model used these results should not be used to examine the

avalanche situation in the study area, but rather for demonstrating the computational performance of Flow-Py. In this example, the dominant term that determines the runout length is the runout angle $\alpha$, but it can also be affected by $Z_{lim}^\delta$. The dominant term that determines the spreading of the process are the divergence ($exp$) and flux cutoff ($R_{stop}$). Combined they can also





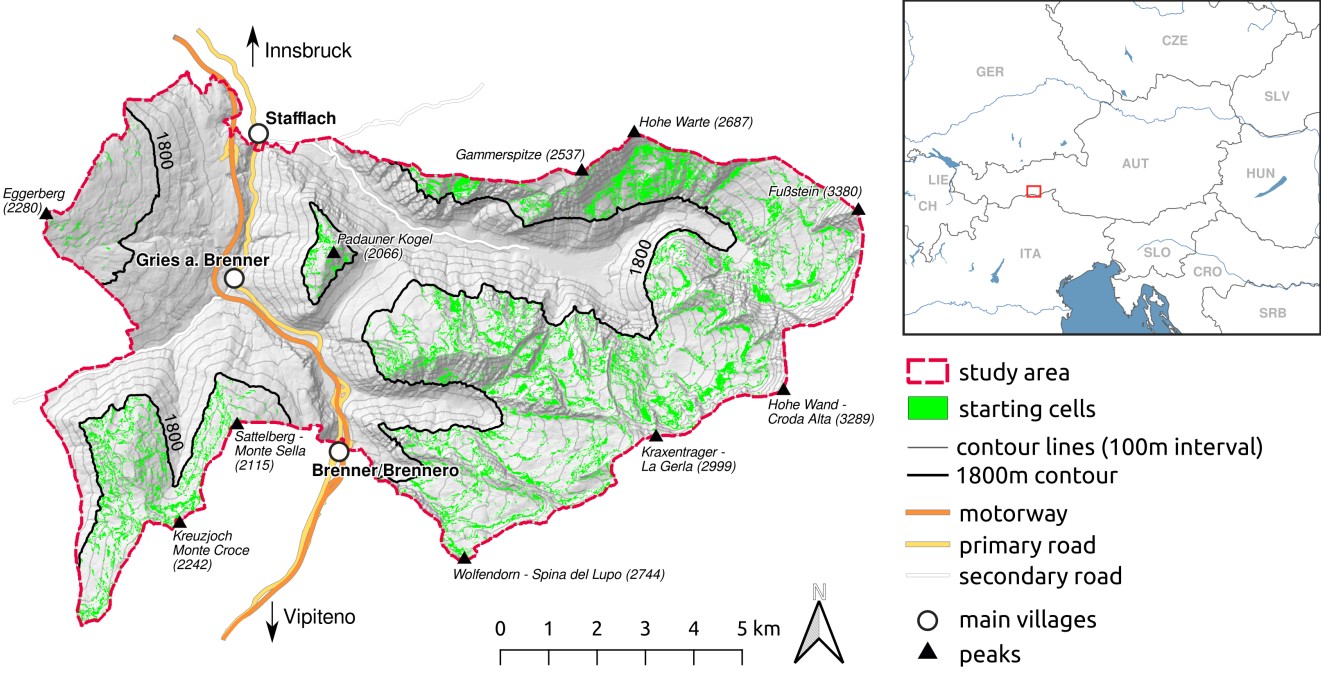

**Figure 7.** Study area for Flow-Py performance testing on a regional scale. Snow avalanche starting cells (green) are defined by locations above 1800 m on slope inclinations between 31 to 34 °. The maps utilize datasets from the following sources: a:©OpenStreetMap contributors 2021. Distributed under the Open Data Commons Open Database License (ODbL) v1.0.; b: Natural Earth. Free vector and raster map data @ naturalearthdata.com; c: Land Tirol - data.tirol.gv.at issued under a Creative Commons Attribution 4.0 International (CC BY 4.0) license.

limit the runout length when $R_{stop}$ is high or divergence (low $exp$-values) is excessively high. The parameterization used in this example has been used in past work for simulations of extreme avalanche events D'Amboise et al. (2021), however there is need for much more extensive parameter studies. In particular the interaction on how $exp$ and $R_{stop}$ interact to limit the spreading of the GMF and the use of $Z_{lim}^{\delta}$ to limit the reach of the GMF.

In our experience the run time of the model is highly dependent on the spreading, and to a lesser extent on the number of starting cells and the length of the runout. Therefore, the feedback that propagates between the routines should not be ignored. A large runout angle (short runout length) will restrict the spreading capabilities even when using a low $exp$ for a highly divergent process.

## 5 Model customization and adaptability

The third computational experiment highlights the adaptability of the Flow-Py simulation tool with an example for a custom extension that was designed to answer a specific question, but additional information and calculations can be easily added into the Flow-Classes (Neuhauser et al., 2021).





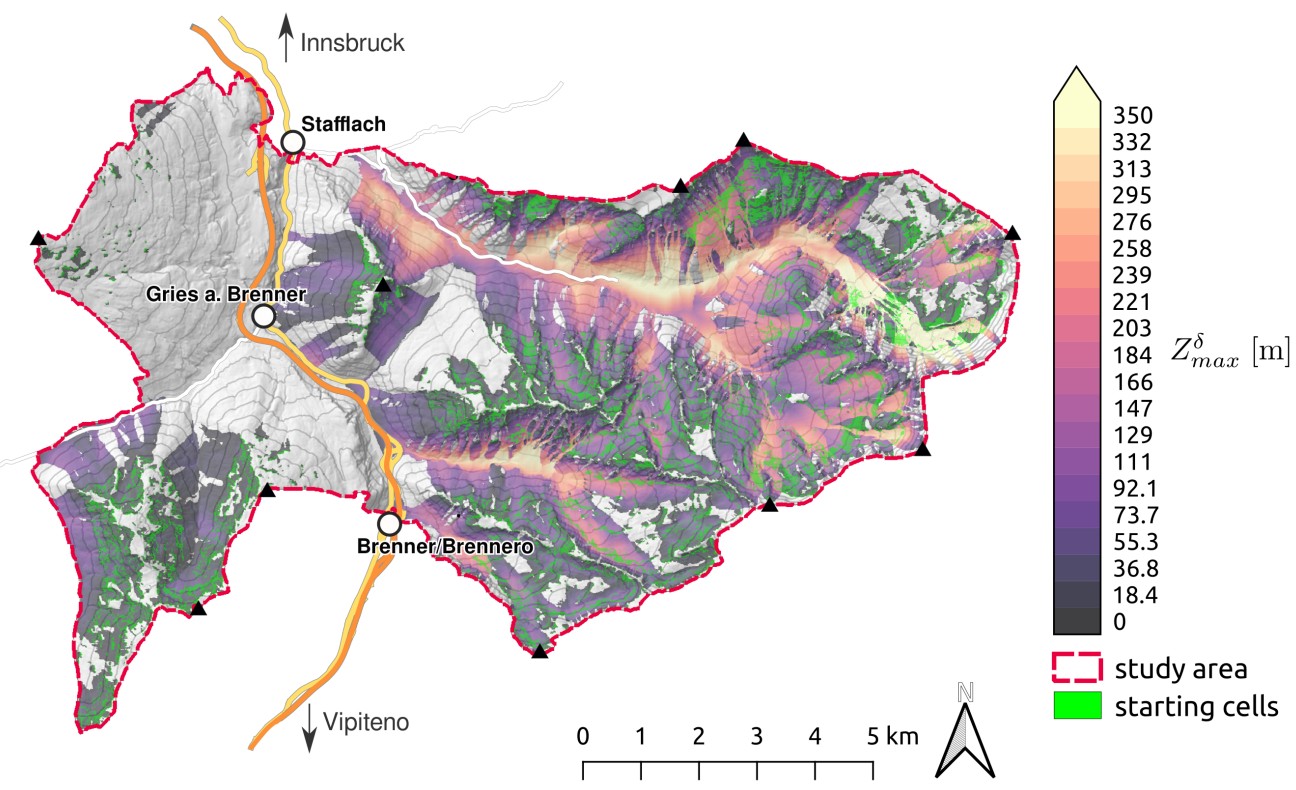

**Figure 8.** Flow-Py simulation results of snow avalanche runout and intensity ($Z_{max}^{\delta}$) in complex terrain on a regional scale. The simulation took 3 h and 45 min. The map utilizes data from ©OpenStreetMap contributors 2021. Distributed under the Open Data Commons Open Database License (ODbL) v1.0. and Land Tirol (data.tirol.gv.at) issued under a Creative Commons Attribution 4.0 International (CC BY 4.0) license.

## 5.1 Experimental setup and methods

This specific model customization experiment addresses the research question: What areas on the terrain are associated with endangering a location containing infrastructure by a GMF? For this example the parabolic slope from Sect. 3.1 and release raster from Sect. 3 were used as input data with the parameterization $\alpha = 25°$, $exp = 8$, $R_{stop} = 3 \cdot 10^{-4}$ and $Z_{lim}^{\delta} = 8,849m$ as well as an additional input raster that contains the location of infrastructure. With the experimental setup defined the initial question can be refined: What raster cells on the synthetic parabolic slope are associated with routing flux of an GMF through a specific set of raster cells that have been identified as locations with infrastructure?





A custom extension called the back-tracking extension was implemented to change the runout model to a model that highlights terrain associated with endangering infrastructure. The back-tracking results will be a spatially explicit subset of the GMF path. Three steps are required to adjust the Flow-Py simulation tool:

1. Load the infrastructure raster as an additional input raster.

2. Adjust the calculation and store new information in the Flow-Class.

3. Save the back-tracking information as a raster and discard the default outputs.

Steps 1 and 3 are simple tasks when using the existing input and outputs of Flow-Py as an example. For the version of Flow-Py used in this contribution an automatic switch was added such that, when an additional input raster is included, Flow-Py will
initiate the back-tracking extension and suppress the normal Flow-Py outputs (ee the Flow-Py repository (Neuhauser et al., 2021) for more information on implementing these steps).

Step 2 is the more challenging adjustment that highlights the adaptable nature of the Flow-Class organization. Since the goal of back-tracking is to find the avalanche starting, transit and runout zones that are associated with endangering infrastructure, a new back-tracking variable must be added to the Flow-Class storing information about a cell's parents. If the back-tracking
variable of a cell is 0, then this cell is not associated with endangering infrastructure; if it is 1, then the cell is associated with endangering infrastructure. After a path is calculated and before updating the result raster, the back-tracking routine can start. Starting with the cells identified as a location with infrastructure a family tree can be constructed by looking at which cells acted as parent cells to these infrastructure cells. For each parent cell the back-tracking variable is changed from 0 to 1. After looping over all cells identified as parent cells that are related to cells containing infrastructure, the result raster can be updated
with the back-tracking results and the next GMF starting cell can be calculated.

To optimize the back-tracking extension with regards to model run time, the starting cells have been ordered by elevation. If a starting cell is located in the path of a previously calculated starting cell at a higher elevation, then the lower elevation starting cell is removed from the queue of starting cells that must be calculated. This will greatly reduce the run time of the model as less starting cells and process paths need to be computed; however, no information about the back-tracking is lost
because the process path of the lower starting cell will be a subset of the upper starting cell's path, but other output raster such as cell counts (CC) and the $Z_{sum}^{\delta}$ are no longer be valid since some starting cells are ignored for optimization.

To test the back-tracking extension two types of infrastructure were considered: a linear infrastructure such as a road, railroad or walking path that crosses the terrain, and a single pixel which could represent a building or utility pole.

### 5.2 Results

Figure 9a shows linear infrastructure (vertical red line) crossing the parabolic terrain, with the areas identified by the back-tracking extension. Most of the path located uphill of the infrastructure has been identified by the back-tracking extension except for few cells that lay on its edges. This is because these cells were not parent cells due to the routing flux-induced stopping criterion.



**Figure 9.** GMF runout modeled with Flow-Py and results of the back-tracking extension with $\alpha = 25°$, $exp = 8$, $R_{stop} = 3 \cdot 10^{-4}$ and $Z^{\delta}_{lim} = 8,849m$. The textured areas highlight starting, transit or runout zones associated with endangering linear infrastructure (a) or a building (b). The topography is a simple parabolic slope connected to a flat plane.





The bottom panel of Fig. 9 shows how the back-tracking extension behaves for a single infrastructure cell (red), e.g. a build-
ing, in the center of the GMF path. A wedge shape subset of the process path starting at the infrastructure cell and extending up
slope is identified by the back-tracking extension, and it is clearly shown that not all uphill cells of the infrastructure cell route
flux through the infrastructure cell. In both linear and building infrastructure cases, all cells that lay below the infrastructure
cells are not identified by the back-tracking extension.

### 5.3 Discussion

The back-tracking extension is a complex adaptation because inputs, outputs and some calculations are changed. However,
because of the modular and adaptive Flow-Py development environment and the advantages of programming in Python's
object-oriented class method, this complex task could be adopted with little effort.

Different routing or stopping approaches could also be easily added to the Flow-Class, which may be necessary to represent
different types of GMFs more precisely. For instance, the additional energy dissipation due to terrain roughness or forest can
be included by accepting different runout angles in the Flow-Class (D'Amboise et al., 2021) as well as a Voellmy-type friction
term (Voellmy, 1955), which is dependent on the GMF intensity ($Z_b^\delta$). Moreover, material flowing versus material sliding or
falling down slope behave differently, which could be described more precisely in the Flow-Py simulation tool by including
more complex routing and stopping routines. However, many of potential Flow-Py extensions will include one or more of the
three steps outlined with the back-tracking examples (i.e. load additional input, adjusting the calculation or save additional
output).

We used the back-tracking extension to exemplify and highlight the adaptability of the Flow-Py simulation tool. By mak-
ing small adaptions Flow-Py was changed from a runout model to a model that identifies endangered infrastructure, which
demonstrates how Flow-Py can be used to investigate questions related to specific GMFs.

### 6 Conclusions and outlook

Flow-Py is an open-source simulation tool for data-based gravitational mass flow (GMF) runout and intensity modeling, which
is suitable for spatially explicit applications on a regional scale. GMF is a term that generalizes the flow of various materials
in different ratios of solids, water (ice) and air down a slope. The GMF behavior, the runout length and the amount of lateral
spreading are all partially dependent on the composition of the material (Pudasaini and Mergili, 2019). Flow-Py handles
diverse flow behaviors by providing an adjustable parameterization that acts to control the spreading and runout lengths of the
simulated GMF path.

Flow-Py's basic model equations and well-organized solver split the GMFs runout modeling into two routines: 1) routing
of the GMF, and 2) stopping of the GMF. The routing routine is further broken down into terrain-based and persistence-based
routing, and the stopping routine is further broken down into two stopping criteria based on runout length and the amount
of flux. With this, Flow-Py provides and educational GMF model development environment, which combines computational
efficiency with low entry barriers for adaptations/extension, such as the presented back-tracking extension.



Besides the local topography two factors influence the spreading of the simulated GMF, namely the $exp$ parameter and the routing flux threshold $R_{stop}$. However, the four main parameters (runout angle $\alpha$, divergence control $exp$, flux cutoff $R_{stop}$ and the limit of the process intensity $Z_{lim}^{\delta}$) have to be defined based on ones experience or corresponding guidelines to obtain the desired range of motion behaviors corresponding to different materials and their compositions. However, further studies

are needed for in depth parameter investigations, including the development of parameter sets which can be used for specific GMF types such as rockfall, different types of snow avalanches or landslides. Part of these parameter studies should include a sensitivity study on the DEM resolution used.

The implementation of the model equations to route the flux on the cell level has the advantage that the flow path does not need to be predetermined in contrast to some similar statistical runout methods (Lied and Bakkehøi, 1980). Therefore, Flow-

Py combines the simplicity of a runout angle-motivated model with the advantages associated with process-based modeling, providing a corresponding intensity measure and allowing for routing in flat or uphill terrain as we demonstrated in a computational experiment. The results of a second experiment show that the run time of the model is suitable for regional modeling (several 100 $km^2$ ) (see Sect. 4). The main factor that control model run time is the parameterization, especially the amount of spreading which is controlled by $exp$ and $R_{stop}$; however, the number of starting cells and the number of available computer

cores are also important factors influencing model run time.

One of the major benefits of Flow-Py compared to existing GMF simulation tools is its well-organized code that allows easy adaptations and extension development. A custom extension was developed for Flow-Py for taking into account terrain complexity with regards to snow avalanches, where automated avalanche terrain exposure scale (ATES) maps were created (Larsen et al., 2020). Future work is being carried out to develop a custom extension which will adapt the stopping criteria to

other statistical models (Lied and Bakkehøi, 1980; Barbolini et al., 2011). We presented the back-tracking extension in Sect. 5 to demonstrate adaptability of the simulation tool, which required adjustments to the input data, calculations and output raster. The additional calculations took advantage of Flow-Py's programming in Python's object-oriented class method, the Flow-Class. The output can be used to identify forests with a direct object protective function by combining the back-tracking results with a map of the current forest cover in a post-processing procedure. More simple extensions have already been developed and

used, e.g., the forest extension which has been applied to quantify the forest's protective effects in transit zones of rockfall and starting, transit and runout zones of snow avalanches, adapting the runout angle stopping criteria dependent on forest structure and the intensity ($Z_{max}^{\delta}$) of the GMF (D'Amboise et al., 2021).

We have shown that Flow-Py is an innovative GMF simulation tool that can be applied for basic simulations (e.g., for hazard zone mapping) but also for more sophisticated custom applications such as identifying areas that potentially endanger specific

infrastructure. Furthermore, not only presenting Flow-Py in this contribution but also the modeling concepts that motivated its model equations and their implementation in the Flow-Py code enables one to reproduce and understand the basic concepts of GMF modeling and to also use Flow-Py as an educational tool.

*Code and data availability.* The Flow-Py code and user manual can be found on thee repository at Neuhauser et al. (2021)



Input data and simulation results can be found at D'Amboise et al. (2021)

*Author contributions.* MN developed the model code with help from JTF and CD. Original draft preparation was done by CD and JTF with significant input from MN (model description) and MT (introduction and conclusions). Figures were designed by AH, MN and CD. Model conceptualization was performed by RF AH AK FP JTF MN and CD with overarching research goals formulated by KK and JTF. Data (DEMs) were obtained/created by JTF and CD. Funding was acuired by FP, MT, JTF and MT. All authors supported the review and editing process of the manuscript.

*Competing interests.* No authors have competing interests.

*Acknowledgements.* This work was conducted in the context of the GreenRisk4ALPs project (ASP635), which has been financed by Interreg Alpine Space, one of the 15 transnational cooperation programs covering the whole of the European Union (EU) in the framework of European Regional policy. Additional financial support from the AvaRange (www.AvaRange.org, international cooperation project "AvaRange - Particle Tracking in Snow Avalanches" supported by the German Research Foundation (DFG, DR 639/22-1) and the Austrian Science Fund
(FWF, I 4274-N29) and the open Avalanche Framework AvaFrame (www.AvaFrame.org, AvaFrame is a cooperation between the Austrian Research Centre for Forests (Bundesforschungszentrum für Wald; BFW) and Austrian Avalanche and Torrent Service (Wildbach- und Lawinenverbauung; WLV) in conjunction with the Federal Ministry Republic of Austria: Agriculture, Regions and Tourism (BMLRT)) projects are greatly acknowledged.



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
