# Peer review of "Flow-Py v1.0: A customizable, open-source simulation tool to estimate runout and intensity of gravitational mass flows"

_Geoscientific Model Development, 2021_

## Author Comment (AC1)

Response 1:

Thank you for taking the time to review our manuscript and the constructive comments. Please find our response to the reviewer's comments below (answers black text, review comments blue italic).

*The paper is an essential contribution to modeling GMFs, especially snow avalanches. During the last couple of decades, most work has been focused on physical models, and most work has been published as proprietary software with high fees to use. These models are still widely used today, but the proprietary software limits scientific development. The Flow-Py tool is a new and improved model built upon existing literature/models. Finally, seeing an open-source model published will be a significant step forward where other researchers can implement and improve the model.*
*Both the paper and the code are well written. I'm able to run the code without any problems.*

Thank you. The main advantages of Flow-Py is certainly it's adaptability allowing for application in science and education – which was one of the main foci.

*One note: several times, it's stated that the code is computationally easy to run. My impression after running the code is that it is relatively computationally heavy to run. The authors do not compare the processing time to similar physical or data-based models, but this is subjective. I'm not sure if, i.e., the RAMMS software for the same 100 km2 extent will take more than the roughly 4 hours mentioned to run the AOI.*

The computational cost of Flow-Py (and any other simulation tool of a similar class) highly depends on the computation example and parameter setting (see e.g. the differences for "lateral spreading" Figure 3 of the manuscript). Computational cost has rarely been noted in the literature, examples include:

- Back tracking with a simulation tool based on Voellmy friction it took up to 18 hours on 1 core to calculate a single avalanche path with r.avaflow, performing 2000 simulation runs (Fischer et al., 2020 )

- Simulations with RAMMS was recently carried out on a 7105 $Km^2$ region in Switzerland using 96 cores (384 GB RAM) taking about 24 hours (Bühler et al., 2022)

- From Rauter et. al (2018) simulations the runout of a single dense flow avalanche of about 2000 m long using OpenFOAM took between several minutes as a $1^{st}$ order interpolation 40m base cell resolution to several hours using a $2^{nd}$ order interpolation.

Because run time is highly dependent on the example (ex. the parameterization, the resolution, etc) and that there are no standardized tests are available we decided to provide a short comparison of Flow-Py and AvaFrame Com1DFA ( Wirbel et. al, 2021) the open source successor of SamosAT ( Sampl and Zwinger 2004) which is of a comparable simulation tool class as RAMMS.

To run the Com1DFA module of AvaFrame release area with corresponding release depth and the digital elevation model are required inputs. To simulate an exceptionally large avalanche with α~25° we use the standard parameter combination, which is recommended to simulate events of catastrophic size. A short experiment on runtime was carried out over a simple parabolic slope seen in Figure 3 in the Manuscript, where the Avaframe com1DFA, shallow water model was run with default parameters (snow depth = 1m) to examine the run time.

A quick comparison to the Avaframe (Wirbel et al., 2021) implementation of Com1DFA took about 70 seconds using the same size release area initialized with 1m of snow and the results are like the Figure 3a (exp = 100). For the low spreading example (Figure 3a in the manuscript) Flow-Py took 1 second to run (the spreading example, Figure 3b took 16 seconds). This run time difference between Flow-Py and Avaframes Com1DFA implementation is between 1 and 2 orders of magnitude. The Avaframe Com1DFA simulations use a 5 m grid which is the default resolution, where Flow-Py calculated on a 10 m grid. It is difficult to project how the run time behavior on a single GMF path will translate to a large domain regional simulation. The Avaframe simulation has been optimized and uses Cython to increase the speed of the simulation where Flow-Py is not. The standard SamosAT version is available in C++ and runs slightly faster than the Cython version. Flow-Py is not optimized for such a run and it does not take full advantage of the parallel processing because there are too few release area cells. Therefore, on larger simulations it is expected that Flow-Py would have an increased run time performance.

See the updated manuscript section 4.3 (paragraph starting on line 417) for updated text on simulation run time.

*How the flux controls the spreading and partly Z is well explained in Figures 3 and 4. In Figure 5, there is an example where the GMF has to overcome a dam in the slope. It would be nice to see an example where the kinetic energy isn't high enough to overcome the dam, and the total runout doesn't reach the given threshold travel angle.*

Thank you for the suggestion. In the following we present a corresponding "dam example". However, we decided not to include the full example in the paper (to avoid the paper becoming lengthy) but to provide the corresponding parameter settings (see lines 351 in the new manuscript, α=30°) such that the interested reader can reproduce the respective simulations themselves.

Within the "dam-example" a run-out angle is chosen such that the avalanche stops on the dam face (α=30°, see figure 1, below). The total run-out is still determined by the threshold of the runout angle, in the presented example close to the crown of the dam, which is uphill from expected runout without dam. Prior routing methods such as the steepest decent would potentially fail while routing in flat and uphill terrain. The dam example highlights the feature that Flow-Py's routing algorithm overcomes this shortcoming.

[Figure]

[Figure]

Fig. 1 GMF runout modeled with Flow-Py on a parabolic slope with a channel and a dam that crosses the terrain at 1350 m with $\alpha = 30°$, $exp = 8$, $Rstop = 3 \cdot 10^{-4}$ and $Z_{\delta lim} = 8{,}849$m. On the upper plot the colors show the value of $Z_\delta$ max which is associated to maximum kinetic energy. The topography is a simple parabolic slope connected to a flat plane. In the lower plot a cross section of the simulation is shown with a comparison of the same simulation with changing $\alpha = 25°$.

*Line 35-36: What is the order of citations? By year (old to new)? Alphabetical? It's not consistent throughout the paper.*

*Minor spelling details:*

*Line 226: "cells"*

*Line 227: Fix degree symbol*

*Line 285: No capital letter in "output"*

*Page 24, footnotes: "thee"*

The line comments have been included and order of citations has been adapted according to the GMD guidelines throughout the text.

References:

Bühler, Y., Bebi, P., Christen, M., Margreth, S., Stoffel, L., Stoffel, A., Marty, C., Schmucki, G., Caviezel, A., Kühne, R., Wohlwend, S., and Bartelt, P.: Automated avalanche hazard indication mapping on state wide scale, Nat. Hazards Earth Syst. Sci. Discuss. [preprint], https://doi.org/10.5194/nhess-2022-11, in review, 2022.

Fischer, Jan-Thomas, Andreas Kofler, Andreas Huber, Wolfgang Fellin, Martin Mergili, and Michael Oberguggenberger. "Bayesian inference in snow avalanche simulation with r. avaflow." *Geosciences* 10, no. 5 (2020): 191.

Rauter, Matthias, Andreas Kofler, Andreas Huber, and Wolfgang Fellin. "faSavageHutterFOAM 1.0: depth-integrated simulation of dense snow avalanches on natural terrain with OpenFOAM." Geoscientific Model Development 11, no. 7 (2018): 2923-2939.

Sampl, Peter, and Thomas Zwinger. "Avalanche simulation with SAMOS." *Annals of glaciology* 38 (2004): 393-398.

Wirbel, A., Oesterle, F., Tonnel, M., and Fischer, J.-T.: avaframe/AvaFrame: Version 0.5, https://doi.org/10.5281/zenodo.5094509, 2021.

---

## Author Comment (AC2)

**Response 2:**

Thanks for taking the time to review our manuscript. Please find our response to the reviewer's comments below (answers black text, review comments blue italic).

**Review 2 summary:**

This paper is an important advancement in the study of GMF and snow avalanches, especially for studies at larger scales and in data space areas. Specifically the work has significant potential to assist with further automation of regional avalanche terrain exposure scale mapping at larger spatial scales. I'm glad to see open-source, object-orientated code allowing for further development and adaptation.

The paper was well written, at the validation methods applied were appropriate. I agree with other commentators that it would also have been interesting to see a comparison of processing times for the same study path using different models. It would have been interesting to see a comparison of the raster output of the model against other GMF approaches, such as TauDEM.

Looking at the literature for TauDEM it is a hydrological tool for terrain based routing. The routing model is the D-infinity multiple flow direction method which allows flow from a parent raster cell to more than one neighbor cell. The TauDEM D-Infinity Avalanche tool has also been adapted for the use of simulating the run out of snow avalanches which uses a simple alpha angle stopping criteria (Tarboton et al., 2015). However, no analysis on the TAUDEM D-Infinity Avalanche performance is available. An in depth model comparison would be interesting but is not within the scope of this work – however the openly available test cases (AVAFRAME, Wirbel et al. 2021) would allow for such an effort. A first step of model performance evaluation (FlowPy, Com1DFA AvaFrame) is described in the answer to reviewer 1.

The code was well documented and there was sufficient guidance to configure the parametrization. I was able to run the code run on both Linux and Windows machines and replicate the sample studies. In practice, I found it was most efficient to conduct simulations over larger spatial scales using an AWS cloud computing implementation to fully benefit from the parallel processing.

Thank you. We would be interested to learn more about the study on the AWS cloud system and are happy that Flow-Py already finds its way into application.

**Technical corrections:**

In 15 - topograhies -> topographies
In 86 - modeling -> modelling
In 399 the these -> these

These corrections were made in the manuscript (We used the American spelling (modeling) throughout the manuscript).

**References:**

Tarboton, David G., Pabitra Dash, and Nazmus Sazib. "TauDEM 5.3: Guide to Using the TauDEM Command Line Functions." (2015).).

Wirbel, A., Oesterle, F., Tonnel, M., and Fischer, J.-T.: avaframe/AvaFrame: Version 0.5, https://doi.org/10.5281/zenodo.5094509, 2021.